# TNFα drives mitochondrial stress in POMC neurons in obesity

Chun-Xia Yi[1,2], Marc Walter[1], Yuanqing Gao[1], Soledad Pitra[3], Beata Legutko[1], Stefanie Kälin[1], Clarita Layritz[1], Cristina García-Cáceres[1], Maximilian Bielohuby[4], Martin Bidlingmaier[4], Stephen C. Woods[5], Alexander Ghanem[6], Karl-Klaus Conzelmann[6], Javier E. Stern[3], Martin Jastroch[1] & Matthias H. Tschöp[1]

Consuming a calorically dense diet stimulates microglial reactivity in the mediobasal hypothalamus (MBH) in association with decreased number of appetite-curbing pro-opiomelanocortin (POMC) neurons; whether the reduction in POMC neuronal function is secondary to the microglial activation is unclear. Here we show that in hypercaloric diet-induced obese mice, persistently activated microglia in the MBH hypersecrete TNFα that in turn stimulate mitochondrial ATP production in POMC neurons, promoting mitochondrial fusion in their neurites, and increasing POMC neuronal firing rates and excitability. Specific disruption of the gene expressions of TNFα downstream signals TNFSF11A or NDUFAB1 in the MBH of diet-induced obese mice reverses mitochondrial elongation and reduces obesity. These data imply that in a hypercaloric environment, persistent elevation of microglial reactivity and consequent TNFα secretion induces mitochondrial stress in POMC neurons that contributes to the development of obesity.

[1] Institute for Diabetes and Obesity, Helmholtz Diabetes Center at Helmholtz Zentrum München, Division of Metabolic Diseases, Department of Medicine, Technische Universität München, German Center for Diabetes Research (DZD), 85764 München-Neuherberg, Germany. [2] Department of Endocrinology and Metabolism, Academic Medical Center, University of Amsterdam, 1105AZ Amsterdam, The Netherlands. [3] Department of Physiology, Augusta University, Augusta, Georgia 30912, USA. [4] Endocrine Research Unit, Klinikum der Ludwig-Maximilians-Universität, 81377 Munich, Germany. [5] Department of Psychiatry and Behavioral Neuroscience, University of Cincinnati, Cincinnati, Ohio 45220, USA. [6] Max von Pettenkofer Institute and Gene Center, Ludwig-Maximilians-Universität, 80539 Munich, Germany. Correspondence and requests for materials should be addressed to M.H.T. (email: tschoep@helmholtz-muenchen.de).

Brain microglia maintain a healthy local environment for optimal neuronal functioning, thereby help ensure innate immune protection under physiological as well as pathophysiological conditions[1]. In response to immune challenges, microglia become activated, causing them to synthesize and release cytokines that consequently trigger pro-inflammatory responses[2]. Depending on the circumstances, microglia-secreted tumour-necrosis factor-$\alpha$ (TNF$\alpha$) and its downstream signals can have beneficial effects on neural survival and protection[3,4], but in other circumstances may exert detrimental effects leading to neuronal dysfunction[5,6]. Consuming a calorically dense (high-carbohydrate, high-fat: HCHF) diet stimulates rapid microglial reactivity in the mediobasal hypothalamus (MBH)[7,8] and is associated with increased TNF$\alpha$ production. When consumption of the HCHF diet becomes chronic and induces obesity (diet-induced obesity; DIO), the induced microglial reactivity persists, and the number of appetite-curbing pro-opiomelanocortin (POMC) neurons decreases[7]. We asked whether the reduction in POMC neuronal function is secondary to the persistent microglial activation with its elevated TNF$\alpha$ production. We found that via stimulating mitochondrial ATP production in POMC neurons, promoting mitochondrial fusion in the neurites and increasing POMC neuronal firing rates and excitability, TNF$\alpha$ induces mitochondrial stress in POMC neurons that in long run, could contribute to POMC neuronal dysfunction in control of energy balance and eventually lead to obesity.

## Results

**Daily rhythm of microglial activity in MBH of lean mice.** Previous studies on the response of MBH microglial reactivity to specific diets were mainly performed during the light phase when nocturnal mice and rats are resting and feeding very little[7,8]. This is important because we have found that during the dark phase (ZT16, 4 h after lights off), the number of iba1-immunoreactive (ir) (activated) microglial cells and of the processes per cell in the MBH were increased relative to levels during the light phase (that is, at ZT4, 4 h after lights on) in lean, chow-fed mice (Fig. 1a,b,e,f). MBH microglial number and activity were associated with elevated gene expression of TNF$\alpha$ (Fig. 1g). In contrast, the gene expression of other cytokines, interleukin (IL)-1$\beta$ and IL-6, did not have this daily-rhythmic pattern (Fig. 1g). The rhythmic pattern of microglial activity was also observed in MBH of rats on chow diet (Supplementary Fig. 1a–d). Fasting chow-fed lean mice for 24 h eliminated the daily-rhythmic pattern of hypothalamic microglial activity (Supplementary Fig. 1e,f,i,j), implying that the pattern is driven by food intake. Furthermore, in comparison to the *ad libitum* condition, 24 h fasting resulted in significantly lower TNF$\alpha$ gene expression, while 4 h refeeding following the 24 h fast caused a significant increase of TNF$\alpha$ gene expression (Supplementary Fig. 1k). We conclude that microglial activity in the MBH represents a physiological response to nutrient intake.

**Microglial activity in MBH in DIO mice elevated persistently.** Importantly, the daily-rhythmic pattern of hypothalamic microglial activity was absent in DIO mice chronically fed the HCHF diet (Fig. 1c–f). The gene expressions of TNF$\alpha$, IL-1$\beta$ and IL-6 were all higher in DIO mice at ZT4 (Fig. 1g), with no further elevation at ZT16. In 24 h fasted DIO mice, the number of iba1-ir microglia was significantly lower at ZT16 than at ZT4 (Supplementary Fig. 1g–j). Thus, in DIO mice fed the HCHF diet, microglial reactivity and TNF$\alpha$ production are persistently elevated throughout the day/night cycle in the MBH, and the local neurons are therefore continuously exposed to high, non-rhythmic levels of TNF$\alpha$.

After 4 months on the HCHF diet, DIO mice had significantly higher numbers of—and more reactive—microglia in close proximity to POMC neurons than occurred in age-matched chow-fed lean mice (Fig. 1h–l). In fact, there was evidence of direct contact between the cell bodies of microglia and neurons (Supplementary Fig. 2a–d). This suggests that in DIO mice, increased inflammatory interactions likely occur directly between POMC neurons and the adjacent reactive microglia, perhaps analogous to the previous finding that toll-like receptor 2-mediated microglial activation also increases the contacts between reactive microglia and POMC neurons[9]. Moreover, consistent with a previous study[7], there was significant POMC neuronal loss after 8 months on the HCHF diet (Fig. 1m).

**TNF$\alpha$ stimulates mito-ATP production in hypothalamic neurons.** Since the reduction of POMC neurons may have been induced by chronic exposure to microglial-derived TNF$\alpha$, and since mitochondria are critical regulators of neurodegenerative pathogenesis[10], we next determined the impact of TNF$\alpha$ on the integrity of mitochondrial bioenergetics in mouse primary hypothalamic neurons, and compared it to the impact of IL-1$\beta$ and IL-6. After 16 h of incubation with 5 nM, but not with 0.5 or 2.5 nM TNF$\alpha$, there was a significant increase of non-mitochondria- and mitochondria-contributed (ATP-linked) cellular oxygen consumption rate (OCR; Supplementary Fig. 3a–d). In contrast, there was no increase of OCR from neurons incubated 16 h with comparable doses of IL-1$\beta$ or IL-6 (Fig. 2a,b and Supplementary Fig. 3e). The TNF$\alpha$-stimulated OCR was paralleled by increased mitochondrial copy number (Fig. 2c)—increased citrate synthase activity (the pacemaker enzyme in the Krebs cycle; Fig. 2d), and increased mitochondrial respiration chain complex I (Supplementary Fig. 3f). Mitochondrial dynamics and bioenergetics are tightly interconnected[11]. Proper mito-fusion has a protective function in that it enables mitochondria to exchange contents for enhanced protein complementation, mitochondrial DNA repair and equal distribution of metabolites[12]. TNF$\alpha$-induced stimulation of complex I was also associated with increased cellular levels of the mito-fusion-regulating protein optic atrophy 1 (Opa1; Supplementary Fig. 4a,b). Therefore, the normal daily rise of TNF$\alpha$ presumably reflects the process of optimally synchronizing neuronal ATP production with hypothalamic cell matrix function, systemic metabolism status and environmental caloric availability.

**TNF$\alpha$ induces mitochondrial elongation in neurites.** To provide sufficient ATP for neural synaptic activity, mitochondria are assembled in soma and then transported along the neurites to reach the synapse. Given that TNF$\alpha$ stimulates Opa1 protein expression, we next determined the impact of TNF$\alpha$ on mitochondrial morphology in neurites of hypothalamic neurons *in vivo*. We generated a G-deleted rabies virus encoding for mitochondrially targeted red fluorescence protein (RABV$\Delta$G-Mito^RFP; Supplementary Fig. 4c). The RABV$\Delta$G-Mito^RFP was injected into the paraventricular nuclei (PVN) unilaterally in lean mice to retrogradely label the MBH–PVN projecting neurons. No detectable reactive microglia or astrocytes were found close to the RABV-infected neurons (Supplementary Fig. 5), indicating no local immune response to RABV-infected neurons. Ten days after the RABV$\Delta$G-Mito^RFP injection, 1 pmol of TNF$\alpha$ was infused into the MBH and mitochondria were assessed 16 h later. In MBH neurons, mitochondria in soma and neurites were labelled by Mito^RFP, and the average length of mitochondria was significantly higher in the TNF$\alpha$-infused group (Fig. 2e–i), demonstrating that TNF$\alpha$ had induced mitochondrial elongation in the neurites of the MBH neurons.

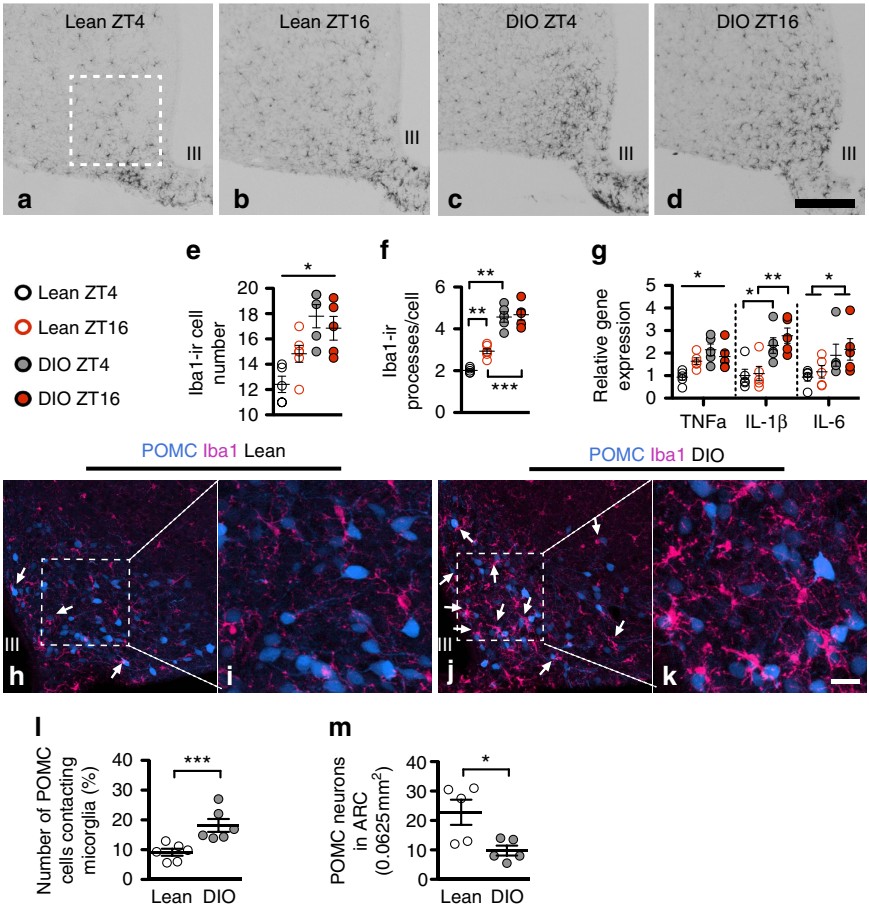

**Figure 1 | Microglial activity in the arcuate nucleus (ARC) persistently elevated in DIO mice with their cell bodies topographically associated with POMC neurons.** (**a–d**) Illustration of iba1-ir cells in the MBH. (**e–g**) For lean and DIO mice, comparing levels at ZT4 and ZT16, there was a significant interaction between time and body weight (BW; lean and DIO) on iba1-ir cell numbers (quantified in a $0.2 \times 0.2$ mm frame outlined by dashed lines in **a** ($n = 5$ for lean ZT4, DIO ZT4 and DIO ZT16; $n = 6$ for lean ZT16, $F_{(2,17)} = 4.504$, $P = 0.049$), and a significant effect of BW on iba1-ir cell numbers at ZT4 ($t_8 = 4.695$, $P < 0.001$ for lean versus DIO at ZT4. For the processes of each microglial cell, there was a significant interaction between time and BW, an effect of BW at ZT4 and ZT16, and an effect of time for lean ($F_{(2,17)} = 4.650$, $P = 0.048$; $t_8 = 9.476$, $P < 0.0001$ for lean versus DIO at ZT4; $t_9 = 6.78$, $P < 0.0001$ for lean versus DIO at ZT16; $t_8 = 3.545$, $P < 0.01$ for ZT4 versus ZT16 in lean mice). There was a significant interaction between time and BW on TNFα gene expression in the MBH ($n = 5$ for each group, $F_{(2,16)} = 5.832$, $P = 0.028$), and an effect of BW on TNFα gene expression at ZT4 ($t_8 = 4.09$, $P < 0.01$). For IL-1β and IL-6, there was only an effect of BW ($t_8 = 2.926$, $P < 0.05$ at ZT4; $t_8 = 3.696$, $P < 0.01$ for ZT16 for IL-β; $F_{(1,16)} = 6.949$, $P = 0.019$ for IL-6). (**h–l**) In DIO mice induced by 4 months of HCHF diet, more iba1-ir microglia were located closely to the POMC cells than in lean mice (identified by *Pomc*-driven eGFP labelling, $n = 6$ mice for lean and for DIO, $P = 0.004$). The contacts between soma and ramified branches of microglia with neuronal soma and neurites are illustrated in **i–k**. (**m**) The number of POMC^eGFP neurons in the ventromedial ARC (vmARC, in a $0.25 \times 0.25$ mm framed area) is significantly reduced in 8-month HCHF diet-fed DIO mice ($n = 5$ for lean or DIO mice, $P = 0.022$). Scale bars, 200 μm in **a–d**; 50 μm in **h,j**; 20 μm in **i,k**. *$P < 0.05$, **$P < 0.01$. Data are presented as means ± s.e.m. $P$ values were analysed by two-way ANOVA followed by Bonferroni multiple comparisons in **e–g**, and by two-tailed Student's *t*-test in **l,m**.

**Mitochondria elongate in neurites of POMC neurons in DIO.** We next injected RABVΔG-Mito^RFP into the PVN of lean and DIO mice to determine whether DIO mice, in which the TNFα level is persistently increased by the reactive microglia, also exhibit elongated mitochondria in POMC neurons. DIO mice had fewer short mitochondria, ranging around 0.5 μm (Fig. 2j–p), but more elongated mitochondria, ranging from 2 to 13 μm (three-dimensional animation of Fig. 2n in Supplementary Video 1). The comparable mitochondrial elongation in neurites of vehicle-treated obese mice and in TNFα-treated lean mice are consistent with TNFα being the cause of mitochondrial elongation.

**TNFα increases POMC neuronal firing rate.** Cellular energy production is driven by the demand for cellular energy consumption. Because neuronal firing is particularly energetically expensive[13], we asked whether TNFα-stimulated neuronal

mitochondrial activity is associated with altered neuronal firing rate and excitability. Patch-clamp electrophysiological recordings were made from identified POMC^eGFP neurons in acute brain slices containing the arcuate nucleus (Fig. 3a–c) and which were pre-incubated either in vehicle or in TNFα. There were no differences in resting membrane potential or input resistance between groups. When continuous firing activity was analysed either in spontaneously active neurons (artificial cerebrospinal fluid (aCSF): 3/9; TNFα: 3/10) or in neurons in which firing was evoked by slight membrane depolarization with d.c. current injection, TNFα caused a significantly higher ongoing firing rate (Fig. 3d–f). Moreover, applying depolarizing current steps of increasing duration resulted in a significantly higher number of evoked action potentials in POMC neurons pre-incubated in TNFα (Fig. 3g–i). Collectively, these results indicate that TNFα drives neuronal energy demands by increasing excitability and

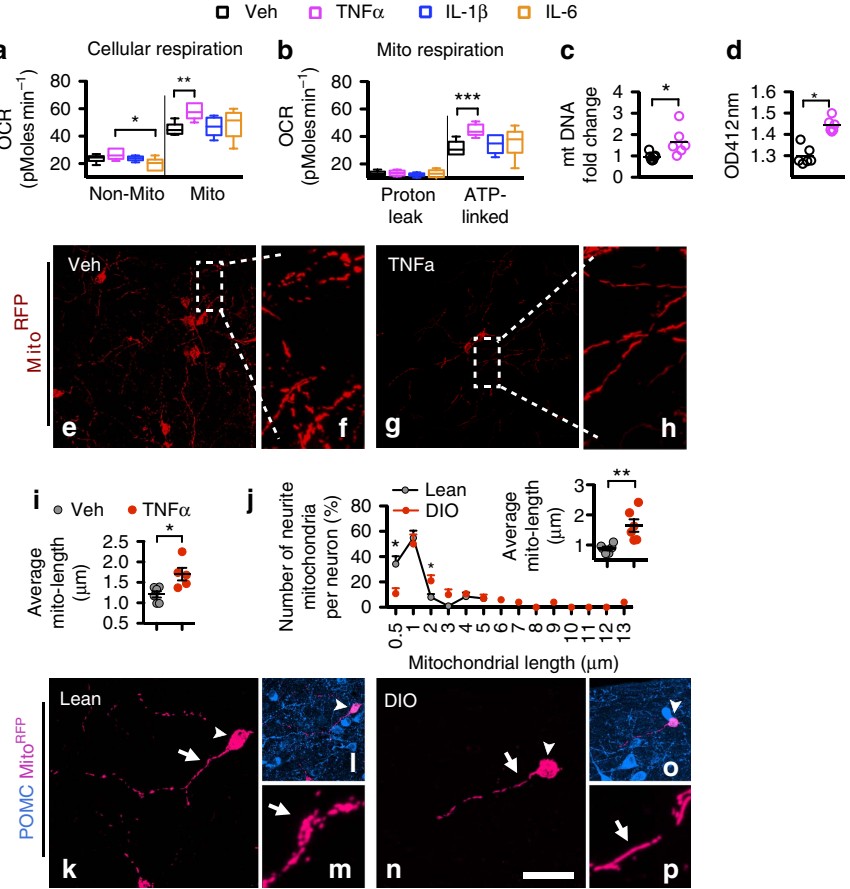

**Figure 2 | TNFα stimulates mitochondrial ATP production and mitochondrial elongation in neurites of hypothalamic neurons.** (**a,b**) Neurons treated with TNFα had a significantly increased ATP-linked mitochondrial OCR in comparison to those treated with vehicle, IL-1β or IL-6 ($n = 6$ wells for each treatment, $F_{(1,20)} = 4.39$, $P = 0.016$ for Mito-OCR, $t_{10} = 4.294$, $P = 0.002$ for TNFα versus vehicle; $F_{(1,20)} = 3.62$, $P = 0.031$ for ATP-linked OCR, $t_{10} = 2.921$, $P = 0.015$ for TNFα versus vehicle); in addition, there was a significant difference in non-mito OCR, $F_{(1,20)} = 4.39$, $P = 0.016$, contributed by the difference between TNFα versus IL-6, $t_{10} = 2.921$, $P = 0.015$). (**c**) Mitochondrial copy number was significantly increased in TNFα-treated neurons ($n = 7$ for vehicle and $n = 6$ for TNFα, $P = 0.018$). (**d**) Mitochondrial citrate synthase activity was unregulated in TNFα-treated neurons ($n = 6$, $P < 0.001$). (**e-i**) Mitochondrial length in TNFα-treated mice in the MBH was significantly longer than in vehicle-injected mice ($n = 6$ for vehicle and $n = 5$ for TNFα, $P = 0.014$). (**j-p**) Mitochondria in the neurites of POMC neurons (arrowheads in **k-p**) were elongated in DIO mice ($n = 6$ mice) in comparison to those in lean mice ($n = 6$ mice; for each length: $P = 0.011$ for 0.5 μm; $P = 0.025$ for 2 μm; $P = 0.058$ for 3 μm; POMC neurons in lean mice did not contain mitochondria longer than 6 μm; for average mito-length, $P = 0.005$), higher magnification of Mito^RFP-labelled mitochondria (arrows) are presented in **l,o**. Scale bars, 100 μm in **e,g**, 12 μm in **f,h**; 30 μm in **k,n**; 60 μm in **l,o**; 8.5 μm in **m,p**. *$P < 0.05$, **$P < 0.01$, ***$P < 0.001$. Data are presented as min to max in **a,b**, and means ± s.e.m. in **c,d,i,j**. $P$ values were analysed by one-way ANOVA followed by *post hoc* $t$-test in **a,b**, and by two-tailed Student's $t$-test in **c,d,i,j**.

that the input/output function of POMC neurons is consequently coordinated with the mitochondrial activity.

**TNFα modulates mito-function in hypothalamic neurons.** To determine the specific molecular underpinnings linking these processes, we next dissected the signalling pathways mediating the effects of TNFα on mitochondrial bioenergetics. Specifically, we profiled key genes involved in mitochondrial respiration by PCR array, and found *Ndufab1* (NADH dehydrogenase (ubiquinone) 1, alpha/beta, subcomplex 1) and *Atp6v1e2* (ATPase, H+ transporting, lysosomal V1 subunit E2) gene expression to be upregulated in response to TNFα (Supplementary Fig. 6a–e). We then used lentiviral particles encoding mouse short hairpin RNA (shRNA) to selectively knockdown receptors of the TNF superfamily (*Tnfrsf*) and their downstream signals (Supplementary Table 1), and subsequently measured OCR in hypothalamic neurons. *In vitro* knockdown of *Tnfrsf11a* and its downstream signalling targets identified the nuclear factor-kB survival pathway to be regulating mitochondrial capacity (Supplementary Fig. 7).

**Knockdown of TNFα downstream signals in MBH reduces DIO.** To further determine TNFα-mediated signalling events in mitochondria, we selectively applied *in vivo* neurotropic serotype 2 adeno-associated virus (AAV-2)-expressing shRNA to knockdown *Tnfrsf11a*, TNF receptor-associated protein1 (*Trap1*, a mitochondrial chaperone that regulates the metabolic switch between mitochondrial respiration and aerobic glycolysis[14]), or *Ndufab1*. Knocking down the gene expression of *Tnfrsf11a*, *Trap1* or *Ndufab1* in the neurons of MBH of lean mice had no effect on food intake or body weight relative to what occurred in response in the AAV-scrambled shRNA control group (Supplementary Fig. 8), indicating that the remaining gene expression (there was 60% knocking-down efficiency) may be sufficient to maintain the functional role of these genes under physiological conditions. In DIO mice, however, targeted gene disruption of *Tnfrsf11a* or *Ndufab1* resulted in significantly reduced food intake and body weight gain (Fig. 4a,b).

**Reverse mito-elongation in POMC neurites in DIO.** RABVΔG-Mito^RFP was then injected into the PVN in combination with

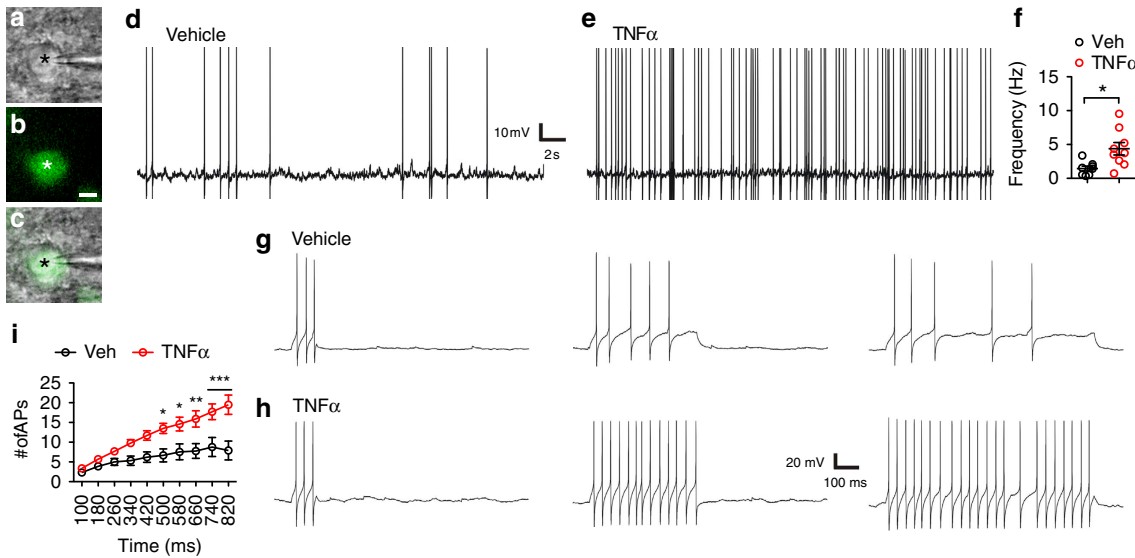

**Figure 3 | TNFα increases neuronal firing activity and evokes excitability of arcuate nucleus POMC neurons.** (a–c) Representative example of a patched POMC^eGFP neuron (asterisk) observed under DIC (a) and fluorescence (b) illumination (a,b superimposed in c). (d,e) Representative traces depicting spontaneous firing activity of two POMC^eGFP neurons that were pre-incubated either in vehicle (aCSF) (d) or in TNFα (e, $1 \mu g \, ml^{-1}$). The mean ongoing firing frequency of POMC^eGFP neurons in each condition is summarized in f ($n = 8$ for vehicle and 9 for TNFα, $P = 0.012$). (g–i) Representative traces of evoked firing potentials (APs) in response to depolarizing steps (25 pA) of increasing durations (100–820 ms) in vehicle (g) and in TNFα (h) ($n = 9$ for vehicle and 10 for TNFα), (i) there was a significant interaction between time and drug on the APs ($F(9,153) = 8.867$, $P < 0.001$), and there were also significant effects of time or drug on APs ($F(1,15) = 41.27$, $P < 0.001$ for time; $F(1,15) = 9.235$, $P = 0.008$ for drug). Scale bar in b, 20 μm. $*P < 0.05$, $**P < 0.01$, $***P < 0.001$, for comparison between vehicle and TNFα groups. Data are presented as means ± s.e.m. $P$ values were analysed by Student's $t$-test (f) or two-way repeated measures ANOVA followed by post hoc $t$-test (i).

AAV-shRNA injection into the MBH to determine whether modulation of mitochondrial dynamics is a possible cause for these effects on systemic energy homeostasis. In comparison to the elongated mitochondria that were present in the control AAV-infused DIO mice, knockdown of *Tnfrsf11a* and *Ndufab1* significantly increased the number of short mitochondria, ranging from 0.5 to 1 μm, and resulted in shorter mean mitochondrial length, respectively (Fig. 4c–j). Thus, in the DIO condition, *Tnfrsf11a*, and to lesser extent *Ndufab1*, but not *Trap1*, are important in mediating the effects of TNFα on mitochondrial dynamics in MBH neurons, as well as in the MBH control of food intake and body weight.

## Discussion

The demand for cellular energy consumption determines cellular energy production. TNFα stimulates the mitochondrial fusion process as well as ATP production, implying that TNFα can drive neurons into the highly energy-demanding state. This is important when there is high synaptic activity and it is characterized by increased neuronal firing rate and excitability, each reflecting the neuronal stress response to TNFα (Supplementary Fig. 9). This concept is supported by the observation that there is adaptively increased maximal respiratory activity in association with increased mito-biogenesis, as well as mitochondrial fusion in the neurites. Under physiological conditions, the day/night cycle of TNFα is coordinated to the resting/feeding cycle, to control cellular energy homeostasis. Furthermore, mitochondrial oxygen consumption is tightly associated with the production of reactive oxygen species[15]. Thus, our findings on TNFα-stimulating POMC neural firing rat are also consistent with the previous report that under physiological conditions, reactive oxygen species activates POMC neurons and reduces feeding[16].

However, in the DIO condition, TNFα levels become constantly elevated, driving persistent neuronal activation and energy demand, and consequently increased mitochondrial activity in neurons. The persistently induced mitochondrial stress ultimately impairs systemic energy homeostasis in key hypothalamic neurons. This increase in mitochondrial and cellular stress will in turn result in more neuronal debris, and thus stimulate microglial activity to achieve debris clearance. Thus, these processes might develop into a vicious circle driving pathological activation of microglia and damaging neuronal function and cellular integrity.

In the current study, the *in vivo* shRNA knockdown of targeted genes were mediated by AAV-2 that dominantly presents natural tropism towards neurons, but not to microglia or other cells in the brain[17], assurances the targeted genes in microglial cells were not affected, thus the effects from *Tnfrsf11a* and *Ndufab1* shRNA on body weight and food intake were not caused by loss of function of the targeted genes in microglia and changes in their daily rhythmic activity.

In summary, we here demonstrate for the first time that microglial TNFα physiologically promotes cellular energy metabolism of key POMC neuron populations that sense and govern systemic metabolism to match nutrient availability. The daily rhythmic pattern of this activity is matched to the diurnal behavioural and food intake patterns of the mice. Overconsumption of calories disrupts the pattern by causing persistently elevated TNFα, which over time causes functional impairment of hypothalamic POMC neurons, thereby generating an additional pathogenic drive towards impaired energy homeostasis and obesity.

## Methods

**Animals.** All rodent studies were approved by and performed according to the guidelines of the Institutional Animal Care and Use Committee of the Helmholtz Center Munich, Germany; the University of Cincinnati; or the Augusta University,

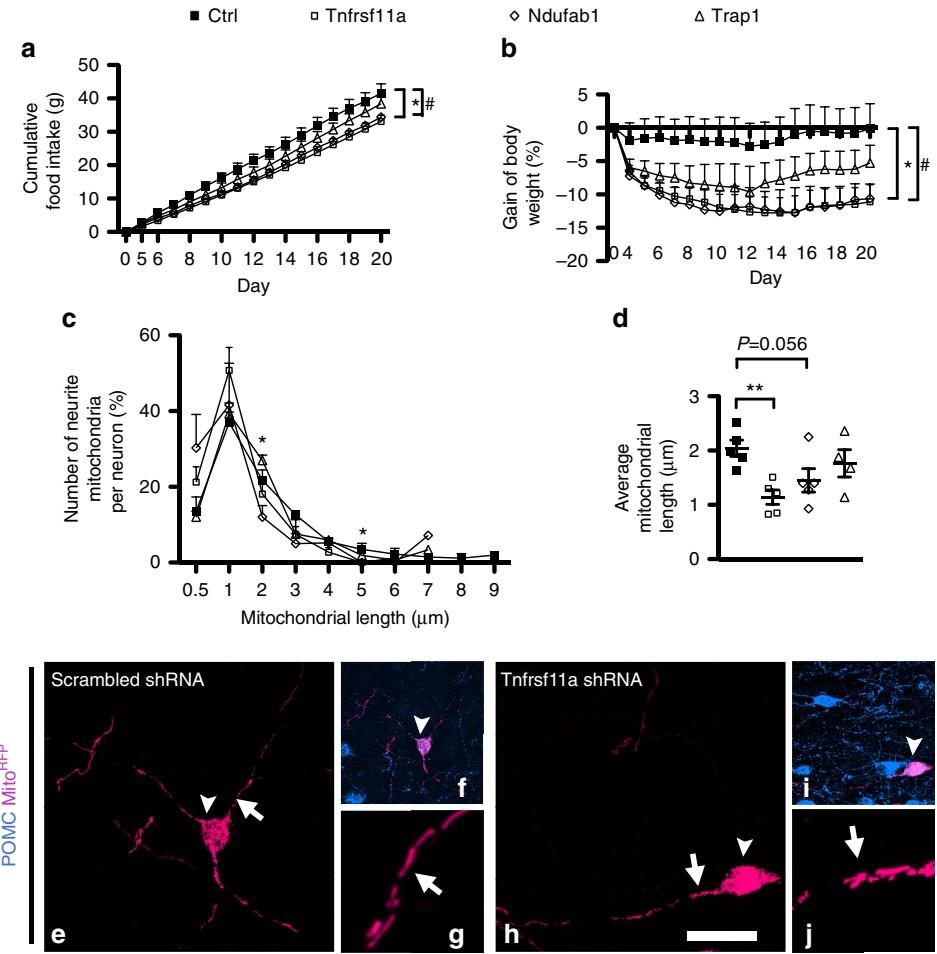

**Figure 4 | Knocking down TNFα downstream signals in the MBH of DIO mice reduces the DIO and reverses the mitochondrial elongation in POMC neurons.** (**a**,**b**) Cumulative food intake and body weight gain at day 20 in DIO mice expressing *Ndufab1* shRNA ($n=8$) or *Tnfrsf11a* shRNA ($n=7$), but not *Trap1* shRNA ($n=10$), were significantly lower than in DIO mice expressing scrambled shRNA ($n=9$; for food intake, $F_{(1,30)}=3.247$, $P=0.036$; $t_{15}=2.437$, $P=0.028$ for *Ndufab1* versus control (Ctrl); $t_{14}=2.204$, $P=0.045$ for *Tnfrsf11a* versus Ctrl; for BW gain, $F_{(1,30)}=2.985$, $P=0.047$, $t_{15}=2.342$; $P=0.034$ for *Ndufab1* versus Ctrl; $t_{14}=2.222$, $P=0.043$ for *Tnfrsf11a* versus Ctrl). (**c**–**j**) Mitochondria in the neurites of POMC neurons (arrowheads in **f**,**i**) of DIO mice expressing *Tnfrsf11a* ($n=5$ mice) or *Ndufab1* shRNA ($n=5$ mice), but not *Trap1* shRNA ($n=4$ mice), were shorter than those in DIO mice expressing scrambled shRNA ($n=5$ mice). For each length, $F_{(1,15)}=4.070$, $P=0.027$ for 2 μm, $F_{(1,15)}=3.468$, $P=0.04$ for 5 μm; for average length, $F_{(1,15)}=4.472$, $P=0.020$; $t_8=4.505$, $P=0.002$ for *Tnfrsf11a* versus Ctrl; $t_8=2.235$, $P=0.056$ for *Ndufab1* versus Ctrl. Higher magnification of Mito$^{RFP}$-labelled mitochondria (arrows) are presented in **g**,**j**. Scale bars, 30 μm in **e**,**h**; 60 μm in **f**,**i**; 8.5 μm in **g**,**j**. *$P<0.05$, **$P<0.01$, ***$P<0.001$. Data are presented as means ± s.e.m. $P$ values were analysed by one-way ANOVA followed by *post hoc t*-test.

USA.

Wild-type male mice (C57BL/6JRj) were purchased from Janvier (Le Genest-Saint-Isle, France); POMC$^{eGFP}$ (Stock No: 009593) and NPY$^{eGFP}$ (Stock No: 006417) male mice were purchased from Jackson Laboratory (Bar Harbor, ME, USA) with the C57BL/6 genetic background. Ten-week-old male Wistar rats were purchased from Charles River (Sulzfeld, Germany). All mice and rats were group housed on a 12 h light, 12 h dark cycle (light from 7:00 to 19:00) at 22 °C, with free access to standard chow diet (LM-485, Teklad) and water before experiments. DIO was induced by HCHF diet purchased from Research Diets (D12331, Research Diets Inc., New Brunswick, NJ, USA). HCHF diet feeding was started at week 8 of age. Body weight and food intake were monitored according to each experimental design. Mice were single-housed after receiving stereotaxic infusions or brain surgeries.

**Primary hypothalamic neuronal culture.** For the primary hypothalamic neuronal culture, hypothalami were extracted from embryonic day 15 (E15) mouse embryos and dissociated to single cells after digestion with trypsin (Sigma-Aldrich, T4674) and DNase I (Invitrogen, 18068-015). Neurons were plated on XF96-PS plates (Seahorse Bioscience, USA) coated with poly-L-lysine (Sigma-Aldrich) at a density of 2,000 neurons per mm$^2$, and they were cultured in Neurobasal (Life Technologies, 12348-017) supplemented with B-27 and GlutaMAX I (Life Technologies, 3505-0061). After 7 days in culture, glial cells were efficiently removed, and neurons started to develop synaptic processes. On day 10, neuronal cells were used for extracellular flux measurement.

**Plate-based respirometry.** To compare the effects of TNFα, IL-1β, IL-6 and vehicle on primary hypothalamic neuronal extracellular flux, cultured primary hypothalamic neurons seeded on XF96-PS plates were treated with TNFα (5 nM, R&D systems, 410-MT, UK), IL-1β (5 nM, R&D systems, 410-ML), IL-6 (5 nM, R&D systems, 406-ML, UK) or 0.5% BSA (vehicle control, Thermo Fisher) for 16 h at 37 °C. Cells were then washed with XF assay medium containing 25 mM glucose (pH adjusted to 7.5) and incubated for 1 h in a 37 °C air incubator. The XF96 plate (Seahorse Bioscience) was then transferred to a temperature-controlled (37 °C) Seahorse (extracellular flux) analyser (Seahorse Bioscience) and subjected to an equilibration period. One assay cycle comprised a 1 min mix, 2 min wait and 3 min measure period. After four basal assay cycles, oligomycin (1 μg ml$^{-1}$) was added by automatic pneumatic injection to inhibit the ATP synthase and thus approximate the proportion of respiration used to drive ATP synthesis and proton leak. After three further assay cycles, carbonyl cyanide-4-(trifluoromethoxy)-phenylhydrazone (0.5 μM) was added the same way to stimulate maximal respiration in mitochondria by chemical uncoupling. After another three assay cycles, rotenone (4 μM) plus antimycin A (2 μM) was added to inhibit the respiratory chain and determine the non-mitochondrial respiratory rate.

**Electrophysiology.** Hypothalamic brain slices were prepared according to the methods previously described[18]. Briefly, male POMC$^{eGFP}$ mice were anaesthetized with pentobarbital (50 mg kg$^{-1}$ intraperitoneal); brains dissected out and hypothalamic coronal slices (210 μm) containing the arcuate nucleus were cut in an

oxygenated ice-cold aCSF, containing in mM: 119 NaCl; 2.5 KCl; 1 MgSO₄; 26 NaHCO₃; 1.25 NaH₂PO₄; 20 D-glucose; 0.4 ascorbic acid; 2 CaCl₂; and 2 pyruvic acid; pH 7.3; 295 mOsm). Slices were placed in a holding chamber containing aCSF, and half slices were either pre-incubated in control aCSF or TNFα ($1\,\mu g\,ml^{-1}$) for 90–120 min at room temperature until use. Slices were transferred to a recording chamber and superfused with continuously bubbled (95% O₂–5% CO₂) aCSF (30–32 °C) at a flow rate of ~3.0 ml min⁻¹. Thin-walled (1.5 mm outer diameter and 1.17 mm inner diameter) borosilicate glass (G150TF-3; Warner Instruments, Sarasota, FL) was used to pull patch pipettes (3–5 MΩ) on a horizontal micropipette puller (P-97; Sutter Instruments, Novato, CA). The internal solution contained the following (in mM): 135 potassium gluconate; 0.2 EGTA; 10 HEPES; 10 KCl; 0.9 MgCl₂; 4 Mg²⁺ATP; 0.3 Na⁺GTP; and 20 phosphocreatine (Na⁺); pH was adjusted to 7.2–7.3 with KOH. Recordings were obtained from fluorescently labelled POMC^eGFP neurons with an Axopatch 200B amplifier (Axon Instruments, Foster City, CA), using a combination of fluorescence illumination and infrared differential interference contrast videomicroscopy. The voltage output was digitized at 16-bit resolution, 10 kHz, and was filtered at 2 kHz (Digidata 1440A; Axon Instruments). Data were discarded if the series resistance was not stable throughout the entire recording (>20% change)[18]. Mean ongoing firing activity and membrane potential values were calculated from a 30 s period using Clampfit (Axon Instruments) or MiniAnalysis (Synaptosoft) software. The input/output function of neurons was assessed by quantifying the number of action potentials evoked in response to consecutive depolarizing pulses of either increased duration (25 pA, 180–820 ms) or increased magnitude (10–90 pA, 100 ms), and plots of the evoked number of action potentials versus step duration or magnitude were generated and compared using two-way repeated measure analysis of variance (ANOVA) followed by Bonferroni's *post hoc* tests.

**Gene expression analysis by real-time PCR.** For gene expression in MBH at ZT4 and ZT16, brain tissues from DIO mice after 10 weeks of HCHF diet or else age-matched lean mice were collected at ZT4 and ZT16, respectively. For gene expression in MBH from the *ad libitum*, fasting and fasting–refeeding studies, brain tissues from *ad libitum*, 24 h fasting (ZT12–ZT12) and 24-h fasting (ZT12–ZT12)–4 h refeeding (ZT12–ZT16) mice were collected at ZT16. MBH was isolated by targeted micropunch. Total RNA was isolated and purified with RNeasy Mini Kit (QIAGEN). cDNA was synthesized by Superscript III 1st Strand Synthesis Kit (Invitrogen). Real-time PCR (RT–PCR) was applied with TaqMan probes (Applied Biosystems). NCBI reference sequences of the target genes for RT–PCR are TNF-alpha: NM013693; IL-1beta: NM008361.3; IL-6: NM031168; and HPRT: NM013556.2. Expression levels of each gene were normalized to housekeeping genes HPRT, relative expression of the targeted gene was presented as fold change compared to the control group that was set at 1.

For the mitochondrial energy metabolism profiling, adult mouse hypothalamic cell line mHypoA-2/12 (CLU177, CELLutions Biosystems Inc.; no mycoplasma contamination was detected) cells were cultured in DMEM supplemented with 10% fetal bovine serum and antibiotics (penicillin 100 IU ml⁻¹ and streptomycin 100 µg ml⁻¹) in 5% CO₂ at 37 °C. Cells cultured in six-well plates with 80–90% confluency were treated either with vehicle (0.5% BSA) or TNFα for 16 h. Cells were snap-frozen in liquid nitrogen for (1) gene expression analysis by Mouse Mitochondrial Energy Metabolism PCR Array (QIAGEN), (2) mitochondrial DNA copy number assay (Detroit R&D, MCN3) and (3) citrate synthase activity colorimetric assay (BioVision, K318-100), according to the manufacturers' instructions.

**ShRNA-mediated gene silencing of TNFα downstream signals in *in vitro*.** CLU177 cells were cultured in DMEM supplemented with 10% fetal bovine serum and antibiotics in 5% CO₂ at 37 °C as formerly described. To evaluate the effects of knocking down the genes along the TNF downstream-signalling pathway, on day 1 30,000 cells were seeded per well in XF96-PS plates. On day 2, the medium was replaced with fresh medium containing 8 µg ml⁻¹ polybrene (hexadimethrine bromide, Sigma-Aldrich, St. Louis, MO). Lentivirus pools containing up to five different shRNA clones (The RNAi Consortium-TRC, Sigma-Aldrich; Supplementary Table 1) per target gene were added to the wells in 2× triplicates. MISSION TRC1.5 non-target shRNA control transduction particles (SHC016V, Sigma-Aldrich) were used as negative control. At 6 h post-lentiviral shRNA transduction the cells were stimulated with TNFα, and 0.5% BSA served as the vehicle control. On day 3 the cellular respiration rates of the treated cells were analysed in an extracellular flux analyser. Data are presented as TNFα-stimulated OCR: $100 \times (OCR_{TNF\alpha} - OCR_{BSA})/OCR_{BSA}$. Two-tailed Student's *t*-test was performed between each targeted gene knocking-down group and non-target shRNA-infected (Ctrl) group.

**Viral construction for *in vivo* infections.** Construction of the monosynaptic glycoprotein (G protein) gene-deleted RABV (SADΔG-GFP) has been described before[19]. To generate RABVΔG-Mito^RFP, a cDNA fragment containing the pre-peptide of human ornithine carbamoyltransferase fused to the N terminus of tagRFP was cloned into the pHHSC-SADΔG-mCherry[20], which allows fast and

reliable virus rescue. The RABVΔG-Mito^RFP was amplified and pseudotyped with the SAD G protein in BSR MG-on cells as previously described[20].

For silencing TNFα downstream signals in *in vivo*, the neurotropic AAV-2 (ref. 17) expresses a scrambled shRNA sequence or sequences for *Tnfrsf11a* (targeting sequence: 5′-GTCCCTGAAATGTGGACCATT-3′), *Ndufab1* (targeting sequence: 5′-GCAGATAAGAAGGATGTGTAT-3′) or TRAP1 (5′-CCGTTATAT TGCTCAGGCTTA-3′), under the control of U6 promoter were customer-made by Vector Biolabs (Philadelphia, USA), and efficiency was tested with ≈60%.

**Stereotaxic infusions into hypothalamus.** Wild-type mice were divided into experimental groups for surgeries, with similar basal body weight and food intake in each group. All surgeries were performed using a mixture of xylazine (5 mg kg⁻¹) and ketamine (75 mg kg⁻¹) diluted in saline as anaesthetic agents, and metamizol (50 mg kg⁻¹, subcutaneous) followed by meloxicam (1 mg kg⁻¹, 3 consecutive days, subcutaneous) was used postoperatively for postoperative analgesia. A motorized stereotaxic system (Neurostar, Tubingen, Germany) was applied for virus injections or cannula implantation.

To compare mitochondrial morphology in neurons in the MBH, the first group of lean mice (age 17–18 weeks) received RABVΔG-Mito^RFP injected unilaterally into the PVN of the hypothalamus, in combination with cannula placement for TNFα or vehicle infusion; the second group of lean or DIO mice (aged 18–19 weeks) only received RABVΔG-Mito^RFP unilaterally into the PVN. A third group of lean or DIO mice (aged 18 weeks) received AAV bilaterally infusion into the MBH to assess their food intake and body weight gain; a fourth group of DIO mice (age 18–20 weeks) received RABVΔG-Mito^RFP unilaterally into PVN, and AAV unilaterally into MBH, to determine the impact of AAV-shRNA on mitochondrial morphology.

For injection into PVN, stereotaxic coordinates were set at −0.8 mm posterior, −0.3 mm lateral to bregma and −4.6 mm depth; for injection into MBH, stereotaxic coordinates were set at −1.6 mm posterior, −0.3 mm lateral to bregma and −5.8 mm depth. After exposing the skull, a small hole was drilled. A 1 µl Hamilton syringe (7000 series, knurled hub) was used for injection of 500 nl of RABV ($7 \times 10^6$ ml⁻¹) or AAV ($1 \times 10^9$ genome copies per ml) per side. Injection speed was set at 50 nl min⁻¹ and the syringe was withdrawn 10 min after each injection.

For infusion of TNFα or vehicle into the MBH in the first group of lean mice that received RABVΔG-Mito^RFP during the same surgery, a second hole was drilled (with the same coordinates for AAV injection described above) on the same side where the RABVΔG-Mito^RFP was injected, and a 22-gauge stainless steel cannula (Plastics One, Roanoke, USA) was implanted and fixed by dental cement.

All mice receiving RABV or AAV injections were monitored for daily food intake and body weight until the end of the study.

In the first group of mice, 10 days after the surgery, 1 pmol of TNFα in 2 µl or 0.5% BSA in 2 µl was infused into the MBH. Mice were killed 16 h later by perfusion fixation as described below. The second and fourth groups of mice were killed by perfusion fixation 10 days after the RABV-Mito^RFP or RABV-Mito^RFP/ AAV injections. The third group of mice was monitored for daily food intake and body weight for 20 days before being killed by perfusion fixation.

**Immunohistochemical and immunofluorescent staining.** For immunohisto-chemical and immunofluorescent staining, mice were killed in CO2 and then rapidly and transcardially perfused with phosphate-buffered saline followed by 4% neutral buffered paraformaldehyde (Thermo Fisher). Brains were extracted, equilibrated in 30% sucrose, sectioned coronally on a cryostat (Leica Biosystems) at 30 µm and collected and rinsed in 0.1 M Tris-buffered saline (TBS).

Immunohistochemical stainings were performed for detecting iba1 from lean or DIO mice (aged 17–19 weeks). Brain sections were incubated with primary antibody of rabbit anti-iba1 (1:1,500, Synaptic Systems, No.234003, Germany). Primary antibodies were incubated overnight at 4 °C; sections were rinsed and incubated in biotinylated secondary goat anti-rabbit IgG (1:400, BA-1000, Vector Laboratories, Inc., Burlingame, CA), sections were then rinsed and incubated in avidin–biotin complex (1:800, PK-4000, Vector Laboratories, Inc., Burlingame, CA) for 1 h and the reaction product was visualized by incubation in 1% diaminobenzidine with 0.01% hydrogen peroxide for 5–7 min. Sections were mounted on gelatin-coated glass slides, dried, dehydrated through a graded ethanol series, cleared in xylene and coverslipped for image collection and quantification. For image acquisition, microscope: Zeiss AXIO Scope A1; objective lenses: Plan-APOCHROMAT ×10/0.45; camera: Zeiss, AxioCam MRc; acquisition software: AxioRel Vision 4.8 were used. Imaging quantification was performed in a blind manner; Iba1-ir cell numbers and processes were manually counted.

Immunofluorescent staining was performed to detect the (1) iba1-ir with enhanced green fluorescent protein (eGFP) in POMC^eGFP mice (aged 16–18 weeks; one section per mouse); or (2) iba1-ir with GFAP-ir in RABVΔG-Mito^RFP-injected mice (one section per mouse); or (3) POMC-ir in RABV-Mito^RFP-injected mice (one section with the most Mito^RFP labelling). Brain sections were incubated with rabbit anti-iba1 (1:1,000), or rabbit anti-iba1 with goat anti-GFAP (1:1,000, Abcam, ab53554), or rabbit anti-POMC (1:1,000, Phoenix Pharmaceuticals, H-029-30) primary antibodies overnight at 4 °C as described above. To detect iba1-ir, sections were rinsed and incubated with biotinylated secondary horse anti-rabbit IgG (1:400, BA-1100, Vector Laboratories) for 1 h, and then rinsed and incubated with streptavidin-conjugated Alexa Fluor 594 (1:300, Jackson ImmunoResearch,

016-580-084, USA) for 1 h. To detect iba1-ir with GFAP-ir, sections were rinsed and incubated with biotinylated secondary horse anti-rabbit IgG for 1 h, and then rinsed and incubated with streptavidin-conjugated Alexa Fluor 488 (1:200, Jackson ImmunoResearch, 016-540-084, USA) and Alexa Fluor 594-conjugated donkey anti-goat IgG (1:200, Jackson ImmunoResearch, 705-166-147) for 1 h. To detect POMC with RFP in RABV-Mito$^{RFP}$-injected brains, sections were rinsed and incubated with biotinylated secondary horse anti-rabbit IgG for 1 h, and then rinsed and incubated with streptavidin-conjugated Alexa Fluor 488 for 1 h. All sections were then rinsed and mounted on gelatin-coated glass slides, dried, covered with polyvinyl alcohol mounting medium containing DABCO (Sigma-Aldrich, USA), observed and imaged by confocal microscopy (Leica SP5, Germany). For image analysis, microscope: Leica TCS SP5 II; objective lenses: HCX PL APO, 63×, numerical aperture = 1.3; imaging medium: glycerol; camera: Leica TCS SP5 II; acquisition software: LAS AF were used. Images were taken by z-stack with sequential scanning of the middle of sections, z-galvo range was 25 μm, z-step sized was 0.1 μm. Images were processed by IMARIS7.6.4 (Bitplane, UK) for three-dimensional reconstruction. For quantification of the number of microglia contacting neighbouring POMC neurons, the total number of eGFP cells and those eGFP cells that had close contact with iba-ir cells were counted, the percentage of contacting cells among the total eGFP cells was calculated in a blind manner. To quantify POMC neuron number in DIO mice fed 8 months of HCHF diet, an additional group of POMC$^{eGFP}$ mice kept under HCHF diet for 8 months was compared to a group of age-matched lean mice. Brain sections were processed with perfusion fixation, sectioning and imaging as described above (one section per mouse). EGFP cell number quantification was performed in a blind manner. Mitochondrial length labelled by RABVΔG-Mito$^{RFP}$ was manually measured by NeuroJ[21] in a blind manner.

**Western blotting.** To detect mitochondrial bioenergetics and dynamics protein expressions in vehicle versus TNFα-treated hypothalamic neurons, CLU177 hypothalamic cells were cultured as described above in six-well plates. At 70–80% confluence, followed by serum starvation for 6 h, cells were treated with TNFα (5 nM) or vehicle (0.5% BSA) for 16 h. After thorough rinsing, cells were snap-frozen in liquid nitrogen. Protein was extracted using RIPA buffer containing protease and phosphatase inhibitor cocktail (Thermo Fisher) 1 mM phenylmethylsulfonyl fluoride and 1 mM sodium butyrate (Sigma-Aldrich, St Louis, MO, USA). The lysates were then centrifuged at 10,000 r.p.m., resolved by SDS–PAGE and transferred to nitrocellulose membranes (45 mmol l$^{-1}$ pore size; Hybond-ECL). After the transfer, the membranes were blocked in 2% blocking reagent provided in the ECL Advance Western Blotting kit (Amersham) for 1 h. Primary mouse anti-complex I (1:1,000, Invitrogen, 438800), mouse anti-complex II (1:5,000, Invitrogen, 458200), mouse anti-complex IV (1:1,000, Invitrogen, 459600), rabbit anti-Opa1 (1:1,000, Abcam, ab42364, USA), mouse anti-mfn2 (1:1,000, Abcam, ab56889, USA), rabbit anti-p-Drp1 (1:1,000, Abcam, ab56788, USA) and rabbit anti-B-actin (1:5,000, Cell Signaling, 4967) antibodies were diluted in TBS with Tween (TBST; pH 7.4) and incubated with the membrane for 1 h at room temperature with gentle rocking. Subsequently, membranes were washed in TBST (three times for 20 min) and incubated with the secondary antibody for 1 h and washed again in TBST (three times for 20 min). Membranes were then developed using the Amersham ECL Advance Western kit.

**Electron microscopy.** Brains were perfused and preserved in 4% paraformaldehyde with 0.2% glutaraldehyde. Mediobasal hypothalamic brain tissue blocks of 1 mm$^3$ were dissected and carefully washed in 0.1 M cacodylate washing buffer. Samples were post-fixed in 1% osmium tetroxide solution and subsequently incubated overnight in 1.5% uranyl acetate solution. Samples were extensively washed with Milli-Q water and then dehydrated with increasing concentrations of ethanol. Samples were then incubated in propylene oxide and later embedded in increasing concentrations of epon mix in propylene oxide. Finally, samples embedded in pure epon were placed in blocks for hardening at 60 °C. Ultrathin sectioning was conducted using an Ultra Microtome Leica UC6. Ultrathin sections of 70 nm were then collected in grids. Grids were contrasted with uranyl acetate and lead citrate before imaging. Images were obtained using a FEI Tecnai T12 transmission electron microscope at 100 kV. Microglia (identified by dense and highly heterochromatin nuclei[22]) and neurons that had contacts were imaged at ×11,000, ×23,000 and ×68,000 magnification for posterior analysis.

**Statistics.** To estimate the number of animals used for in vivo studies and number of samples for in vitro studies, we have used data from our recent similar studies for power calculations. Furthermore, Shapiro–Wilk tests for normality for previous data sets revealed normal distribution. Overall, we regarded P values <0.05 as statistically significant. All data are expressed as means ± s.e.m. Statistical analyses were performed using Prism5.0 (GraphPad). Two tailed Student t's-test, one-way or two-way ANOVA or two-way repeated measure ANOVA followed by post hoc analysis was used to test for differences between individual experimental groups. Group size estimations were based on similar studies published previously.

**Data availability.** The data that support the findings of this study are available from the corresponding author on reasonable request.

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

## Acknowledgements

C.-X.Y. is supported by an AMC fellowship and Dutch Diabetes Fonds (2015.82.1826), the Netherlands. A.G. and K.-K.C. are supported by Sonderforschungsbereich (SFB)870, Germany. J.E.S. is supported by a National Institute of Health (NIH) grant (R01 HL090948), USA. M.H.T. is supported by the Alexander von Humboldt Foundation, the Helmholtz Alliance ICEMED and the Helmholtz Initiative on Personalized Medicine iMed by Helmholtz Association, Helmholtz cross-programme topic 'Metabolic Dysfunction', the German Research Foundation DFG (SFB1123 & Nutripathos Project ANR- 15-CE14-0030) as well as the European Research Council ERC (AdG HypoFlam no. 695054).

## Author contributions

C.-X.Y. conceptualized and co-designed the project, supervised the experiments, performed in vivo viral injections and morphology analyses, interpreted the findings and drafted the manuscript; M.W. performed in vitro gene silencing and mito-biogenesis measurements; Y.G. performed western blot and RT–PCR analysis; S.P. performed the electrophysiology study; B.L. and C.G.-C. performed cell cultures; C.L. performed extracellular flux measurement; M.B. contributed to performed rats study; M.B. conceived the project; S.C.W. drafted the manuscript; A.G. and K.-K.C. generated RABV; J.E.S. supervised the electrophysiology study; M.J. drafted the manuscript; M.H.T. conceptualized and co-designed all experiments; interpreted the findings and drafted the manuscript.

## Additional information

**Competing interests:** The authors declare no competing financial interests.

