## [Peer Review File · Nature Communications]

Reviewers' comments:

Reviewer #1 (Remarks to the Author):

This is an interesting study identifying a relationship between hypothalamic microglia and POMC neurons. A nice hypothesis is put forward for a role of microglial TNF- α and mitochondrial events in POMC neurons in response to high fat diet exposure. I am overall supportive of this paper.

However, it was surprising to see the limited amount of references cited. More curious is the fact that no contemporary hypothalamic works are cited that revolve around mitochondria, cellular stress, ROS etc. That needs to be rectified. There is a very recent paper on hypothalamic microglia-POMC interplay (<http://www.ncbi.nlm.nih.gov/pubmed/27405276>), which the authors should cite also, and, discuss how those results can be put in perspective of the present study.

Reviewer #2 (Remarks to the Author):

The current manuscript by Yi and colleagues demonstrates how microglial TNF α contributes to the damage in POMC neurons and leads to obesity.

The authors found that TNF α challenge could induce mitochondrial ATP production and mitochondrial elongation in neurites of cultured hypothalamic neurons and proposed that microglial TNF α was responsible for these physiological changes in-vivo. They showed activated microglia and increased neuronal mitochondrial fusion in obese mice. Through gene knockdown of TNF α signaling protein, the oxygen consuming rate of hypothalamic neurons was reduced in-vitro, and mitochondrial elongation and increased food intake in DIO mice were rescued in-vivo. These findings propose that regulation of cytokine level inside the body may exert therapeutic efficacy in preventing or/and reversing the obesity in human patients.

The current manuscript is at high standard and the hypothesis is clearly addressed by the experiments. However, the authors should provide additional data and information to improve further the manuscript.

1. The authors found that the diurnal pattern of hypothalamic microglia was impaired in DIO mice while they found that microglial activity of DIO rats decreased at ZT16 as compared to those at ZT4. Is there any explanation for this finding? Have the authors measured the protein expression of cytokines by western blot or ELISA in these DIO rats to see whether the cytokine pathway was related to this change?
2. How did the authors count the number of POMC neurons contacting with neighboring microglia? How authors randomized the results and how many cells they counted per animal? Ultrastructural studies using lectronic microscopy should be performed to confirm direct contacts, and determine whether they are functional (i.e. showing the presence of clathrin-coated vesicles or other trans-endocytosis mechanisms, etc.).
3. Although the authors found that only gene expression of TNF α increased at ZT16 of chow fed mice, they still need to elaborate more on why they focused on investigating the TNF α pathway since they also found increases in IL1 β and IL6 gene expressions in DIO mice. As a consequence, TNF α might not be necessarily related to obesity, or not represent the main player involved.
4. What is the concentration of IL6 used for the plate-based respirometry study?
5. As TNF α can mediate neuronal survival and death through differential binding to different TNF receptors depending on the treatment time and dose, have the authors performed dose-response studies by respirometry of TNF α to investigate whether the same changes would be observed? The same question is also applied to IL1 β .
6. The authors showed that only one mito-fusion regulating protein was induced by TNF α treatment in cultured hypothalamic neurons. Have the authors performed western blot on hypothalamic tissue from DIO mice to determine whether additional mito-fusion regulating

proteins are stimulated under obesity?

7. The authors should perform ultrastructural studies by electronic microscopy to assess mitochondrial fusion as it is difficult to see the changes of mitochondrial morphology inside neuronal bodies using RABVΔG-MitoRFP injection. Also, multiple mitochondria might overlap one with each other in the neuronal processes, which might bias quantification.
8. Have the authors studied mitochondrial fission as another indicator to say whether mitochondria were responding to the stress induced by TNFα?
9. Knockdown of *Tnfrsf6* significantly increased the OCR in the cultured neurons. Does it represent a potentially protective effect of TNFα when binding to this receptor? Is there any previous report on this receptor?
10. The authors saw different responses to food intake and body weight between lean and DIO mice under targeted gene disruption of TNFα signaling molecules. Have the authors checked whether the basal food intake activity was the same between lean and DIO mice? Or the gene expression rates of these TNFα signaling molecules were different between two mice?
11. The authors concluded that the increase in mitochondrial and cellular stress by TNFα will in turn result in more neuronal debris to stimulate microglial activity to achieve debris clearance. Are there any references to suggest that TNFα specifically leads to the degeneration of POMC neurons in MBH? Or was neurodegeneration induced by multiple factors upon high calorie diet?
12. Were there functional experiments performed (such as electrophysiology or retrograde dye labeling) to conclude that neuronal function is impaired by high calorie diet induced by TNFα? That would be important to strengthen the current findings.
13. How did the authors confirm that neuronal damage was driven only by microglial TNFα but not by peripheral (e.g. monocyte, natural killer cell)-derived TNFα under high calorie diet? Have they measured the peripheral levels of TNFα in DIO mice to identify the primary source of TNFα?

Reviewer #3 (Remarks to the Author):

The report by Yi et al. builds on earlier observations that a hypercaloric diet causes an inflammatory response in the mediobasal hypothalamus by implicating mitochondrial changes in POMC neurons as a contributor to obesity. Findings are novel and of potentially broad interest. However, several of the proposed mechanisms are not sufficiently supported by the data and require additional experimentation. For instance, while the group convincingly demonstrates the involvement of a particular TNFα receptor, the role that the reported mitochondrial changes play in vivo is unclear. Furthermore, lack of specificity of adenoviral knockdown for POMC neurons is a major shortcoming of the report that precludes my recommendation for publication. Specific critiques:

1. The authors suggest a direct link between TNFα signaling and the observed mitochondrial phenotypes, which they suggest are evidence of "mitochondrial stress". Instead, they should consider the increased mitochondrial connectivity and ATP production as an adaptive response to increased cellular stress. Indeed, stress-induced mitochondrial hyperfusion was previously reported to increase ATP production and protect mitochondria from mitophagy. Is it possible that mitochondrial adaptations occur indirectly as a consequence of increased firing of POMC neurons? More in vitro experiments are required to examine underlying mechanisms further.
2. The knockdown experiments shown in Fig. 3 implicate one of the TNFα receptors as important in food intake and weight gain, but add little additional insight into the role of mitochondria in POMC neurons. In particular, the authors seem to suggest that upregulation of *ndufab1* is somehow a bad thing that contributes to hyperphagy in obesity. Knocking down an essential ETC subunit in POMC, AgRP, and other cells is akin to a MBH lesion experiment and adds no information beyond that, in toto this brain region promotes food intake.
3. A selective upregulation of complex I is suggested as the mechanism by which TNFα increases respiratory capacity. However, the OXPHOS antibody cocktail detects a single complex I subunit

only, and in the absence of precedent, it is difficult to image how an induced imbalance of ETC proteins could improve mitochondrial function. The PCR array data show that many ETC components trend towards upregulation. Increased mitochondrial biogenesis should therefore be examined as an alternative mechanism.

4. The authors posit a “day/night cycle of TNF α [that] is coordinated to the resting/feeding cycle” and refer to “diurnal”, even “circadian” (implying circadian clock) regulation of microglial activation and TNF α secretion. However, the data as it stands are as easily explained by the fed state of the animals. This needs to be addressed better, for instance by limiting access to food to the light period, or by gavage-feeding starved animals after examining microglial activation at different time points.

5. Opa1 has roles unrelated to mitochondrial fusion and, given that inner and outer mitochondrial membrane fusion are coordinated, it is not apparent how upregulation of a protein that mediates fusion of the inner mitochondrial membrane only could promote mitochondrial elongation. Oxidation of Mfn1/2 due to glutathione depletion and other modifications implicated in stress-induced mitochondrial hyperfusion are more likely mechanisms.

Other concerns:

That “disruption of specific TNF downstream signals” reduces obesity (abstract and elsewhere) is an overstatement, since only KD of Tnfrs11a was tested.

Fig. 2m does not show signs of POMC neuron loss (l. 73)

The statement “More fusion processes are required to move mitochondria than to sustain stationary mitochondria” is not supported by the literature. Indeed, mitochondrial fission is generally thought to be critical for axonal/dendritic transport of mitochondria.

ShRNA-mediated knockdown is not “targeted gene disruption”.

Reviewer #1 (Remarks to the Author):

This is an interesting study identifying a relationship between hypothalamic microglia and POMC neurons. A nice hypothesis is put forward for a role of microglial TNF- α and mitochondrial events in POMC neurons in response to high fat diet exposure. I am overall supportive of this paper.

However, it was surprising to see the limited amount of references cited. More curious is the fact that no contemporary hypothalamic works are cited that revolve around mitochondria, cellular stress, ROS etc. That needs to be rectified. There is a very recent paper on hypothalamic microglia-POMC interplay (<http://www.ncbi.nlm.nih.gov/pubmed/27405276>), which the authors should cite also, and, discuss how those results can be put in perspective of the present study.

We thank the reviewer for being enthusiastic about our findings and for making us aware of these opportunities to improve our referencing. Indeed, our study on the microglia-neurons interaction is consistent with other data reported in PMID: 27405276 "Hypothalamic TLR2 triggers sickness behavior via a microglia-neuronal axis", where the authors showed that toll-like receptor 2 induces microglia occupation of POMC cells and increases the rate of glutamatergic innervation, thus affecting synaptic input organization. Our novel observations that activated microglia migrate closer to POMC neurons in response to nutrient intake are consistent with that innate immunity based concept. Together, these findings jointly add up to a transformative new model at the core of which reside dynamic microglia-neuron interactions regulating hypothalamic neuro-circuits in control of energy homeostasis. Moreover, in our revision, we present novel data from electrophysiological studies dissecting the stimulatory impact of TNF α on POMC neural firing rates and excitability. Together with our previously presented results on the stimulatory effects of TNF α on mitochondrial oxygen consumption rates, both data sets are in line with the previous discovery that under physiological conditions, mitochondria derived ROS can activate POMC neurons (PMID: 21873987 "Peroxisome proliferation-associated control of reactive oxygen species sets melanocortin tone and feeding in diet-induced obesity" and PMID: 14561818 "Mitochondrial formation of reactive oxygen species"). As suggested, we now added discussion of these points in the revision.

Reviewer #2 (Remarks to the Author):

The current manuscript by Yi and colleagues demonstrates how microglial TNF α contributes to the damage in POMC neurons and leads to obesity.

The authors found that TNF α challenge could induce mitochondrial ATP production and mitochondrial elongation in neurites of cultured hypothalamic neurons and proposed that microglial TNF α was responsible for these physiological changes in-vivo. They showed activated microglia and increased neuronal mitochondrial fusion in obese mice. Through gene knockdown of TNF α signaling protein, the oxygen consuming rate of hypothalamic neurons was reduced in-vitro, and mitochondrial elongation and increased food intake in DIO mice were rescued in-vivo. These findings propose that regulation of cytokine level inside the body may exert therapeutic efficacy in preventing or/and reversing the obesity in human patients.

The current manuscript is at high standard and the hypothesis is clearly addressed by the experiments. However, the authors should provide additional data and information to improve further the manuscript.

1. The authors found that the diurnal pattern of hypothalamic microglia was impaired in DIO mice while they found that microglial activity of DIO rats decreased at ZT16 as compared to those at ZT4. Is there any explanation for this finding? Have the authors measured the protein expression

of cytokines by western blot or ELISA in these DIO rats to see whether the cytokine pathway was related to this change?

We thank the reviewer for these kind and supportive comments. In Supplementary Fig.1e-j, we show that in food deprived (fasting) DIO mice, the iba1-ir microglial cell number was decreased at ZT16 compared to ZT6, but it was still significantly higher than in lean mice at both ZT4 and ZT16. It was technically not straightforward feasible to generate meaningful data on specific microglial protein production of cytokines in mouse hypothalami by western blot or ELISA. We did make serious efforts over the recent months to measure TNF α protein in mouse hypothalamic tissue, but so far did not manage to establish reliable readouts. Challenges may potentially be due to the fact that the TNF α protein concentration in homogenous hypothalamic tissue is very low as a consequence of dilution from the distinct cell populations that are producing TNF. Our interpretation of these data is that although microglial activity is governed by feeding, there is also intrinsic biological clock machinery inside the microglia (similar to the liver biological clock machinery, that feeding regulated metabolic gene expression largely override the intrinsic biological clock controlled metabolic gene expression). Both regulatory influences modulate microglia activity in a complex overlay. In support of that model, we do find that TNF α regulates canonical clock gene expression programs. DIO mice kept on HCHF diet exhibit impaired day-night feeding pattern, therefore the intrinsic biological clock machinery inside the microglia in DIO mice appears to differ from that of lean mice. Because after 24 hours of food deprivation, regulatory effects of the intrinsic biological clock, still remain even though effects of nutrients are absent, some level of difference of microglial activity can still be detected at some of the time points.

2. How did the authors count the number of POMC neurons contacting with neighboring microglia? How authors randomized the results and how many cells they counted per animal? Ultrastructural studies using electronic microscopy should be performed to confirm direct contacts, and determine whether they are functional (i.e. showing the presence of clathrin-coated vesicles or other trans-endocytosis mechanisms, etc.).

To count the number of POMC neurons contacting with neighboring microglia, we selected one section from each POMC^{eGFP} mouse that received chow or HCHF diet. Sections were chosen with 30 μ m thickness. We then stained for iba1, and fluorescent images were taken for iba1-ir and eGFP. The total number of eGFP cells, and those eGFP cells that had close contact with iba-ir cells were counted (since two cells are very close to each other, we can see overlapping of red and green fluorescence (i.e. yellow) in most of the cases), and percentage of contacting cells among the total eGFP cells was calculated. We have now added this quantification method to the Methods section.

We agree with the reviewer that electron microscopy is a proper approach to confirm direct contacts of microglia and POMC neurons. To study the ultrastructure of the microglia-neuron contacts, we therefore now have performed electron microscopy studies in chow or HCHF diet fed wild-type mice (since we known that in the arcuate nucleus, the majority of the neurons that are close to microglia in DIO mice are POMC neuron, we looked closely in these cells). We found that wherever microglia was in proximity to neurons, there were cellular membrane attachments between the microglial cell soma and neuron soma. We now added image material depicting such contacts in the revised version (Supplementary Fig.2a-d).

3. Although the authors found that only gene expression of TNF α increased at ZT16 of chow fed mice, they still need to elaborate more on why they focused on investigating the TNF α pathway since they also found increases in IL1 β and IL6 gene expressions in DIO mice. As a consequence, TNF α might not be necessarily related to obesity, or not represent the main player involved.

We thank the reviewer for raising this important issue. The multiple functions of TNF α and its complex receptor signaling systems in physiology and pathology remain highly diverse and are

still not clear. Our focus on TNF α does of course not exclude the possible roles played by other cytokines. In the present study, we chose to study TNF α due to the fact that only TNF α gene expression rises and falls under physiological condition, while that of other microglial produced cytokines like IL1 β and IL6 only rises in DIO condition. This indicates the role of TNF α might switch between physiology and pathology during the HCHF diet induced metabolic disorder. Given the important roles of TNF we have now identified, we indeed intend to dissect the role of other microglial cytokines with their downstream pathways in the hypothalamus and their potential link to the neural control of energy metabolism.

4. What is the concentration of IL6 used for the plate-based respirometry study?

The concentration of IL6 used for the plate-based respirometry study is also 5nM, the same as the TNF α and IL1 β concentrations.

5. As TNF α can mediate neuronal survival and death through differential binding to different TNF receptors depending on the treatment time and dose, have the authors performed dose-response studies by respirometry of TNF α to investigate whether the same changes would be observed? The same question is also applied to IL1 β .

This is a very good and important point. We indeed have performed dose-response studies of TNF α with 0.5nM, 2.5nM and 5nM at an early stage of the study. All three doses can mildly increase the non-mitochondrial respiration in comparison to vehicle control, but mitochondrial respiration/ATP-linked mitochondrial respiration reached significant difference in cells treated with 5nM TNF compared to vehicle control. We therefore then chose 5nM TNF α for the follow-up experiments. Since we did not observe clear effects of IL1 β and IL6 on mitochondrial respiration/ATP-linked mitochondrial respiration, we did not perform dose response studies on these two cytokines with lower doses. We now have added the dose response data into the revision (Supplementary Fig.3).

6. The authors showed that only one mito-fusion regulating protein was induced by TNF α treatment in cultured hypothalamic neurons. Have the authors performed western blot on hypothalamic tissue from DIO mice to determine whether additional mito-fusion regulating proteins are stimulated under obesity?

We thank the reviewer for this question. We have not performed western blots on hypothalamic tissue from DIO mice to check mito-fusion protein expression. The main reason is that the hypothalamus contains highly heterogeneous cell populations and that is therefore difficult to isolate cell specific changes in single key proteins. We assume that even if we observed changes of mito-fusion proteins from in vivo animal models, we cannot exclude that such a phenomenon might be causally relevant for changes of mitochondria in astrocytes, microglia, oligodendrocytes, endothelia or tanycytes.

7. The authors should perform ultrastructural studies by electronic microscopy to assess mitochondrial fusion as it is difficult to see the changes of mitochondrial morphology inside neuronal bodies using RABV Δ G-MitoRFP injection. Also, multiple mitochondria might overlap one with each other in the neuronal processes, which might bias quantification.

We thank the reviewer for the comment and this suggestion. The aim of the current study is to understand whether and how the mitochondria trafficking inside the neurites between the soma and synapse is affected by hypercaloric environments. It is difficult to answer this question by electron microscopy, as we know, we therefore generated the unique RABV Δ G-MitoRFP to tackle this issue.

We have studied mitochondrial morphology in cell bodies of RABV Δ G-MitoRFP infected neurons, we also performed electron microscopy and compared the mitochondria morphology between the RABV Δ G-MitoRFP methods and electron microscopic method. In RABV Δ G-MitoRFP infected

neurons, we found that under physiological conditions, mitochondrial diameter and length vary from less than 0.5 μ m to 5 μ m (please see in Supplementary Video.1), while in electron microscopy, we could hardly find any 5 μ m mitochondria. This suggests there is a limitation for use of electron microscopy method to study the absolute length of mitochondria in neurons. We also found the mitochondrial morphology inside the soma can be completely different from those in the neurites, for example, when mitochondria are highly fragmented or “beagle”, the mitochondria in distal of the neurites can be elongated considerably (please see the reference data 1, file “RABV Δ G-MitoRFP-neuron”). We are currently working on solving the technical issues on imaging the RABV Δ G-MitoRFP infected neurons by super-resolution microscopy (Leica TCS SP8 STED), by which we can visualize the semi-ultrastructure of the MitoRFP labeled mitochondria. We expect to be able to stain mitochondrial bioenergetics and dynamics markers in the MitoRFP labeled mitochondria, as an alternative method than electron microscopy to study neuronal mitochondrial function and morphology.

In other parallel projects, we have also studied the mitochondrial trafficking in the neurites in cultured primary hypothalamic neurons (co-cultured with astrocytes). We labeled the mitochondria by mito-tracker green dye and obtained time-lapse movies in single neurites. In majority of the neurites, there are always bi-directional two tracks, one track for transporting mitochondria from the soma to the nerve terminal, and the other track from the terminal back to the soma (please see the reference data 2, file “Primary neuron astrocytes co-culture mitotracker green”). In the current *in vivo* study, we also observed such two tracks in most of the neurons we studied. And the MitoRFP labeled mitochondria in two tracks can be clearly separated, without overlapping. Nevertheless, we understand the concerns of the reviewer, therefore, as we have mentioned, ultimately, the ideal quantification should be done by super-resolution microscopy. This is something we started to work on but will unfortunately not be able to deliver shortly for these models and questions due to technical challenges described above.

8. Have the authors studied mitochondrial fission as another indicator to say whether mitochondria were responding to the stress induced by TNF α ?

We thank the reviewer for the comments. We have compared mitochondrial fission/fusion protein expression in vehicle versus TNF α treated hypothalamic neurons, and only found a significantly increase of OPA1 protein expression, no changes were found in mitofusion2 and mitochondrial fission phosphor-Drp1 protein expression (Supplementary Fig.3a and b).

9. Knockdown of Tnfrsf6 significantly increased the OCR in the cultured neurons. Does it represent a potentially protective effect of TNF α when binding to this receptor? Is there any previous report on this receptor?

We thank the reviewer for raising this interesting issue. Tnfrsf6 is also called the FAS receptor. It is a so-called death receptor on the cell surface that can lead to apoptosis. Our data indicate that via the Tnfrsf6 pathway, TNF α could exert inhibitory effects on OCR that could lead to a decrease of ATP production, which might lead to cell death. While the TNF downstream pathway is a complex, and Tnfrsf6 would not be the only pathway responding to TNF α in hypothalamic neurons. Our OCR data indicated that if the Tnfrsf6 pathway mediates the inhibitory effect on OCR, this pathway on OCR might be not dominant in the TNF α treated hypothalamic neurons, since in our study, the overall effects of TNF α is stimulating OCR. On the other hand, this observation also illustrates again that TNF α might induce different types of pathophysiological responses via different pathways under different circumstances.

10. The authors saw different responses to food intake and body weight between lean and DIO mice under targeted gene disruption of TNF α signaling molecules. Have the authors checked whether the basal food intake activity was the same between lean and DIO mice? Or the gene expression rates of these TNF α signaling molecules were different between two mice?

We thank the reviewer for the question. In our study, the lean mice were kept on chow diet, and

DIO mice were on high carbohydrate high fat (HCHF) diet, the basal level of the daily food intake between these lean and DIO mice are: 4.04 ± 0.07 g chow diet vs. 3.06 ± 0.11 g HCHF diet, i.e. 15.51 ± 0.27 kcal chow diet vs. 16.99 ± 0.63 kcal HCHF diet, with different nutrients components. The selection of the targeted gene for shRNA knocking down was based on gene expression array studies in cultured hypothalamic neurons, since so far there is no ideal approach to isolate hypothalamic neurons from adult brain tissue and profile gene expression without losing some of the functional and cellular integrity of the cell. We speculate the differing responses to food intake and body weight between lean and DIO mice were due to the different capacities in maintaining neuro-circuits in regulating energy metabolism, with the remaining gene expression that is not completely knocked down, hypothalamic neuro-circuits of lean mice were capable of managing metabolic control, while in HCHF diet induced DIO mice, the remaining gene expression is not sufficient to handle the metabolic stress caused by HCHF diet.

11. The authors concluded that the increase in mitochondrial and cellular stress by TNF α will in turn result in more neuronal debris to stimulate microglial activity to achieve debris clearance. Are there any references to suggest that TNF α specifically leads to the degeneration of POMC neurons in MBH? Or was neurodegeneration induced by multiple factors upon high calorie diet?

We thank the reviewer for this question. For the microglia-neuron interaction under hypercaloric environment, we propose a “vicious circle hypothesis”. The impact of high calorie diet might be complex, in one of our studies (currently under review elsewhere), we found that HCHF diet can specifically induce accumulation of advanced glycation end products (AGEs) in POMC and NPY neurons, but not in microglia or astrocytes, while the receptor for the AGEs are not expressed by neurons, but highly expressed by microglia (as well as pericytes and endothelial cells). This could indicate that the AGEs might be secreted as waste products by neurons and taken up by microglia, indeed we found AGEs can stimulate microglial TNF α gene expression. In the current study, we found microglial TNF α can in turn acts on neurons, and in the long run, through the mitochondrial mechanism, lead to neuronal dysfunction. We propose that such neuron-microglia interactions under hypercaloric environments forms a “vicious circle”, in which the neural derived waste requires microglial activation as a physiological response, while persistent microglial activation then however eventually exerts detrimental effects on those specific neurons the microglia was initially intended to support.

12. Were there functional experiments performed (such as electrophysiology or retrograde dye labeling) to conclude that neuronal function is impaired by high calorie diet induced by TNF α ? That would be important to strengthen the current findings.

We agree with the reviewer that more functional experiments are needed to understand how the neuronal function is impaired by high calorie diet induced by TNF α . We therefore performed electrophysiological studies on the effects of TNF α on POMC neuronal activity. This study was performed in POMC^{eGFP} mouse brain slices. We pre-incubated slices with aCSF or TNF α (90-120 min) and measured firing activity and membrane potential. We found that TNF α increased excitability and input/output function of POMC neurons. Although these neuronal activities were recorded in acute TNF α treating condition, we proved that the energy demands caused by TNF α stimulated neuronal excitabilities could be the driving force of mitochondrial activity, as we state in the final discussion “The demand for cellular energy consumption determines cellular energy production”. Such process can also take place under physiological conditions in the lean mouse hypothalamus, where a day/night cycle of TNF α is coordinated to the resting/feeding cycle, to regulate the cellular energy homeostasis. In the diet-induced obese condition, the TNF α remains constantly elevated, which drives persistent neuronal activation, energy demand and thus mitochondrial stress in neurons, and which will eventually cause neuronal dysfunction. We have now added these important and mechanistic data into an independent figure (Figure.3).

13. How did the authors confirm that neuronal damage was driven only by microglial TNF α but not by peripheral (e.g. monocyte, natural killer cell)-derived TNF α under high calorie diet? Have they measured the peripheral levels of TNF α in DIO mice to identify the primary source of TNF α ?

We agree with the reviewer that we cannot entirely exclude the possibility that peripheral monocytes and NK cell derived TNF α could act in the CNS, since it is known that TNF α can be transported from blood to brain in mouse. For the peripheral levels of TNF α , earlier pioneers of metabolic disease research as for example, Gökhan Hotamisligi et al, found that in ob/ob and db/db mice, serum TNF α level does increase comparing to lean mice, however, the overall level of TNF α in circulation level is very low, that even in obese mice, detectable readouts could be only be found in about half the mice examined. We also have attempted to measure TNF α protein by ELISA (Mouse Cytokine Antibody Array, R&D) in both brain tissue and serum, with maximal hypothalamic brain tissue or serum that we can afford. However, we got very poor readout for TNF α , while some other cytokines and chemokines such as CXCL13 (BCA-1), CD54, M-CSF, CCL2 and CCL17 showed clearer readouts. Therefore, so far we can only assume that there is elevation of peripheral levels of TNF α in DIO mice. Primarily, TNF α is an autocrine cytokine to stimulate microglia activity, therefore, if there is periphery infiltrated TNF α , it might also act on microglia to stimulate their activity. Eventually, such a process would result in higher concentration of TNF α in the microenvironment and act on neighboring neurons to affect neuronal function.

Reviewer #3 (Remarks to the Author):

The report by Yi et al. builds on earlier observations that a hypercaloric diet causes an inflammatory response in the mediobasal hypothalamus by implicating mitochondrial changes in POMC neurons as a contributor to obesity. Findings are novel and of potentially broad interest. However, several of the proposed mechanisms are not sufficiently supported by the data and require additional experimentation. For instance, while the group convincingly demonstrates the involvement of a particular TNF α receptor, the role that the reported mitochondrial changes play in vivo is unclear. Furthermore, lack of specificity of adenoviral knockdown for POMC neurons is a major shortcoming of the report that precludes my recommendation for publication. Specific critiques:

1. The authors suggest a direct link between TNF α signaling and the observed mitochondrial phenotypes, which they suggest are evidence of "mitochondrial stress". Instead, they should consider the increased mitochondrial connectivity and ATP production as an adaptive response to increased cellular stress. Indeed, stress-induced mitochondrial hyperfusion was previously reported to increase ATP production and protect mitochondria from mitophagy. Is it possible that mitochondrial adaptations occur indirectly as a consequence of increased firing of POMC neurons? More in vitro experiments are required to examine underlying mechanisms further.

We agree with the reviewer that the mitochondrial adaptations might occur indirectly as a consequence of increased firing of POMC neurons following stimulation by TNF α . We therefore now performed electrophysiology studies on POMC neurons, examining firing rate and excitability following TNF α treatment. We found that TNF α increases excitability and input/output function of POMC neurons. Although these neuronal activities were recorded in acute settings following TNF α treatment, we proved that the energy demands caused by TNF α induced stimulation of neuronal excitabilities could be the driving force of mitochondrial activity, as we state in the final discussion "The demand for cellular energy consumption determines cellular energy production". Such process could take place under physiological conditions in the lean mouse hypothalamus, where a day/night cycle of TNF α is coordinated with the resting/feeding cycle, to regulate the acute demands of cellular energy homeostasis. This day/night cycle of TNF α was mimicked by our *in vitro* studies with relatively short TNF α treatment (2hours to 16 hours). Under these conditions, we have observed what most likely are adaptive responses. If these adaptive responses take places along the day-night cycle, it would mean that when hypothalamic TNF α is low, the adaptive response involving ATP production and mito-fusion is also lower, which would mean the microglia-neuron interaction would stay within the physiological range. However, in high-fat high-sugar diet induced obese conditions, when TNF α level is constantly elevated, such adaptive response will remain high along day-night cycle, and the sustained "non-resting"

mitochondrial activity may result in bioenergetic deficiency and altered dynamics, which will impair function of those specific neurons. We have now added these substantial data (Figure.3) and would like to use the opportunity to thank the reviewer for the suggestion to generate and add such data.

2. The knockdown experiments shown in Fig. 3 implicate one of the TNF α receptors as important in food intake and weight gain, but add little additional insight into the role of mitochondria in POMC neurons. In particular, the authors seem to suggest that upregulation of *ndufab1* is somehow a bad thing that contributes to hyperphagy in obesity. Knocking down an essential ETC subunit in POMC, AgRP, and other cells is akin to a MBH lesion experiment and adds no information beyond that, in toto this brain region promotes food intake.

We agree with the reviewer that if the complex-1 is fully knocked down, we could expect a severe cellular dysfunction as Complex 1 has an essential role in maintaining mitochondrial function and integrity. However, the actual knock down is targeting a subunit of the complex 1, and the efficiency of the knock down was calibrated around 60%. This means we have generated and observed the consequences of partially removing a subunit of complex 1. We believe that this is a reasonable approach to testing functional contribution of this downstream pathway given all limitations including the ones rightfully pointed out by the reviewer.

3. A selective upregulation of complex I is suggested as the mechanism by which TNF α increases respiratory capacity. However, the OXPHOS antibody cocktail detects a single complex I subunit only, and in the absence of precedent, it is difficult to image how an induced imbalance of ETC proteins could improve mitochondrial function. The PCR array data show that many ETC components trend towards upregulation. Increased mitochondrial biogenesis should therefore be examined as an alternative mechanism.

We agree with the reviewer on this point and therefore have performed experiments to measure mitochondrial biogenesis. We found the total mitochondrial copy number as well as the activity of the Krebs cycle pacemaker enzyme citrate synthase were both upregulated by TNF α stimulation (Fig.2c and d). This upregulation of mitochondrial biogenesis may reflect the cumulative effects from each individual ETC component. The detected change on *Ndufab1* might indicate that this subunit may play a more important role than the other complex 1 subunits.

4. The authors posit a “day/night cycle of TNF α [that] is coordinated to the resting/feeding cycle” and refer to “diurnal”, even “circadian” (implying circadian clock) regulation of microglial activation and TNF α secretion. However, the data as it stands are as easily explained by the fed state of the animals. This needs to be addressed better, for instance by limiting access to food to the light period, or by gavage-feeding starved animals after examining microglial activation at different time points.

We thank the reviewer for this comment. In our fasting experiment (supplementary Fig.1e-j), we show that fasting eliminates the day-night differences of the microglial activity. We therefore concluded the day/night cycle of TNF α is coordinated coupled to the resting/feeding cycle. To further address this issue, following the suggestion of the reviewer, we now performed a fasting-refeeding experiment in chow diet fed mice. We fasted mice from ZT12 (right before the dark period) for 24 hours, and then refeed mice from the beginning of the dark phase for 4 hours. Mice were sacrificed at ZT16 and hypothalamic TNF α , IL1 β and IL6 gene expression were measured. We found that in comparison to the *ad libitum* mice, 24 hours fasting exhibited significantly lower TNF α gene expression, and a trend toward lower IL1 β gene expression. Refeeding for 4 hours led to significantly increased TNF α gene expression, and trended towards higher IL1 β gene expression, in comparison to the fasting group. These data confirm that nutrient intake is one of the main driving forces of TNF α gene expression and microglial activity. We have now added these data into the Supplementary Fig.1k, and replaced the words “diurnal” and “circadian” to be “daily (-fluctuating)”.

5. Opa1 has roles unrelated to mitochondrial fusion and, given that inner and outer mitochondrial membrane fusion are coordinated, it is not apparent how upregulation of a protein that mediates fusion of the inner mitochondrial membrane only could promote mitochondrial elongation. Oxidation of Mfn1/2 due to glutathione depletion and other modifications implicated in stress-induced mitochondrial hyperfusion are more likely mechanisms.

We thank the reviewer for this comment. We agree that the detection of changes in Opa1 might be a mere association, rather than the key mediator of TNF α 's regulatory impact onto mito-fusion. As we have mentioned in the response to Reviewer 2 (question 7), mitochondrial morphology can differ substantially between soma and neuritis. This means that regulatory control over mitochondrial dynamics may also be different between soma and neurites, after all, the trafficking of mitochondria along the neurites might require another complex transporting system (for example the kinesin superfamily motor proteins). In our study, the analyses of protein level were all performed in whole cell homogenous samples, which does not allow for separation of soma from neuritis. This issue could mask the actual protein expression level in each cell compartment. We have now modified the description of the opa1 data in the revision following the reviewer's guidance.

Other concerns:

That "disruption of specific TNF downstream signals" reduces obesity (abstract and elsewhere) is an overstatement, since only KD of Tnfrsf11a was tested.

We thank the reviewer for pointing this out. We have changed this statement to "Disruption of specific TNF downstream signals Tnfrsf11a or Ndofab1".

Fig. 2m does not show signs of POMC neuron loss (l. 73)

We thank the reviewer pointing out this error. We apologize for the mistake, it is indeed Fig.1m we meant here.

The statement "More fusion processes are required to move mitochondria that to sustain stationary mitochondria" is not supported by the literature. Indeed, mitochondrial fission is generally thought to be critical for axonal/dendritic transport of mitochondria.

We thank the reviewer for pointing this out. Since we do not have real mito-motion data, and as the reviewer has suggested, that the increased mitochondrial connectivity and ATP production are adaptive responses to increased cellular stress, we now removed this statement.

ShRNA-mediated knockdown is not "targeted gene disruption".

We thank the reviewer for pointing this out to us, we have now corrected this to be "Knock down of gene expression".

REVIEWERS' COMMENTS:

Reviewer #2 (Remarks to the Author):

The revision addresses most of my concerns. However, a few elements would still require clarification:

In remark 2: The wording within the methods section is still ambiguous about which observers were blinded for what analysis. Were the observers unaware of the diabetic status of the animals?

In figure 1m, it would be important to provide the density of POMC neurons (number per surface area), instead of their overall number. Additionally, is analysing one 30 micron section sufficient to accurately determine the density of POMC neurons in the vmARC? I would recommend to provide the number of POMC neurons that were counted for each animal in the methods.

The response to remark 6, stating an inability to determine cell-type-specific reactions to TNFalpha, raises concerns regarding microglial-specific production of TNFalpha leading directly to neuronal responses. The knockdown of TNF signaling pathways in figure 4 is interesting, but is TNFRs11a specific to neurons, or is it also expressed on other cell types? Does this knockdown have an effect on microglial diurnal response within the region?

In response to remark 11, the authors cite the "vicious cycle hypothesis" but more evidence should be provided, especially given the lack of citations for/against TNFa having a specific degeneration of POMC neurons in the MBH.

The addition of figure 3's electrophysiology data is helpful, but the authors do not link it back to DIO. Is there relevant literature about the firing patterns of POMC neurons from animals with DIO? It should be cited here.

Reviewer #3 (Remarks to the Author):

The new data, especially the electrophysiology, greatly strengthen the impact of the study.

REVIEWERS' COMMENTS:

Reviewer #2 (Remarks to the Author):

The revision addresses most of my concerns. However, a few elements would still require clarification:

In remark 2: The wording within the methods section is still ambiguous about which observers were blinded for what analysis. Were the observers unaware of the diabetic status of the animals?

The quantification of the POMC^{eGFP} neurons that had close contact with iba-ir cells were counted blinded as same as for the POMC^{eGFP} neurons in mouse that received chow or HCHF diet, we now added this into the revision.

In figure 1m, it would be important to provide the density of POMC neurons (number per surface area), instead of their overall number. Additionally, is analysing one 30 micron section sufficient to accurately determine the density of POMC neurons in the vmARC? I would recommend to provide the number of POMC neurons that were counted for each animal in the methods.

To be sure that the comparison of POMC neuronal number will be made at the same level of the hypothalamus, we have collected the brain section from the same level of the hypothalamus of each mouse, The quantified of the number of POMC neurons was performed in a fixed frame of $250\ \mu\text{m} * 250\ \mu\text{m} = 0.0625\text{mm}^2$, we now added this into the revision. Since these sections are very precious and were shared by other project, we are unfortunately not able to provide data on the total POMC neurons per brain of each animal.

The response to remark 6, stating an inability to determine cell-type-specific reactions to TNFalpha, raises concerns regarding microglial-specific production of TNFalpha leading directly to neuronal responses. The knockdown of TNF signaling pathways in figure 4 is interesting, but is TNFRs11a specific to neurons, or is it also expressed on other cell types? Does this knockdown have an effect on microglial diurnal response within the region?

The adeno-associated virus (AAV) we have used is serotype 2. AAV2 dominantly presents natural tropism towards neurons, but not to microglia or other cells in the brain (Watakabe A et al. *Neurosci Res*, 2015), we do not expect the knockdown had an effect on microglia and their diurnal activity. We now specified this neurotropism of AAV2 in the Method of "Viral construction for *in vivo* infections", and added this in the Discussion.

In response to remark 11, the authors cite the "vicious cycle hypothesis" but more evidence should be provided, especially given the lack of citations for/against TNFa having a specific degeneration of POMC neurons in the MBH.

We agree with the reviewer that we do need more evidence to support the "vicious cycle hypothesis". However, this is out of the scope of the current study.

The addition of figure 3's electrophysiology data is helpful, but the authors do not link it back to DIO. Is there relevant literature about the firing patterns of POMC neurons from animals with DIO? It should be cited here.

So far, most of the electrophysiological studies on POMC neuronal activity were

performed by pharmacological or genetic approaches, no data are available for comparing the firing pattern of POMC neurons specifically between lean and DIO animals. Our data on the impact of TNF α on POMC firing rate suggest that the energy demands from the neuronal activity drive the energy production upon TNF α stimulation. We assume that under DIO condition, there might be a persistent TNF α stimulation of POMC neurons during day and night, thus such stimulation of firing rate should occur during day and night.

Reviewer #3 (Remarks to the Author):

The new data, especially the electrophysiology, greatly strengthen the impact of the study.